# How do multi-morbidity and polypharmacy affect general practice attendance and referral rates? A retrospective analysis of consultations

**Andrew O'Regan**[1]*, **Jane O'Doherty**[1], **Ray O'Connor**[1], **Walter Cullen**[2], **Vikram Niranjan**[3], **Liam Glynn**[1,4], **Ailish Hannigan**[1]

**1** School of Medicine, Health Research Institute, Faculty of Education and Health Sciences, University of Limerick, Limerick, Ireland, **2** School of Medicine, Health Sciences Centre, University College Dublin, Dublin, Ireland, **3** School of Public Health Physiotherapy and Sports Science, University College Dublin, Dublin, Ireland, **4** Health Research Board Primary Care Clinical Trial Network, Galway, Ireland

\* andrew.oregan@ul.ie

## Abstract

### Background

As prevalence of multimorbidity and polypharmacy rise, health care systems must respond to these challenges. Data is needed from general practice regarding the impact of age, number of chronic illnesses and medications on specific metrics of healthcare utilisation.

### Methods

This was a retrospective study of general practices in a university-affiliated education and research network, consisting of 72 practices. Records from a random sample of 100 patients aged 50 years and over who attended each participating practice in the previous two years were analysed. Through manual record searching, data were collected on patient demographics, number of chronic illnesses and medications, numbers of attendances to the general practitioner (GP), practice nurse, home visits and referrals to a hospital doctor. Attendance and referral rates were expressed per person-years for each demographic variable and the ratio of attendance to referral rate was also calculated.

### Results

Of the 72 practices invited to participate, 68 (94%) accepted, providing complete data on a total of 6603 patients' records and 89,667 consultations with the GP or practice nurse; 50.1% of patients had been referred to hospital in the previous two years. The attendance rate to general practice was 4.94 per person per year and the referral rate to the hospital was 0.6 per person per year, giving a ratio of over eight attendances for every referral. Increasing age, number of chronic illnesses and number of medications were associated with increased attendance rates to the GP and practice nurse and home visits but did not significantly increase the ratio of attendance to referral rate.

**Data Availability Statement:** Sharing of this dataset is restricted by the Irish College of General Practitioners research ethics committee. Access to

the dataset is available upon reasonable request to the Irish College of General Practitioners Research Ethics Committee (research@icgp.ie).

**Funding:** The authors received no specific funding for this work.

**Competing interests:** The authors have declared that no competing interests exist.

## Discussion

As age, morbidity and number of medications rise, so too do all types of consultations in general practice. However, the rate of referral remains relatively stable. General practice must be supported to provide person centred care to an ageing population with rising rates of multi-morbidity and polypharmacy.

## Background

Increasing age, multi-morbidity and polypharmacy present important challenges to health care systems and general practice has a front-line role in their management [1, 2]. However, general practice in Britain has been described as being 'in crisis' for several years due to extremely serious challenges with recruitment and workload [3, 4]. In England, a recent funding strategy will support Primary Care Networks, whereby practices work collaboratively within a locality with a population health agenda, and it remains to be seen if this can successfully alleviate the pressure on this key component of the health service [5]. Studies of databases generated from millions of general practitioner (GP) consultations have shown significant increases in consultation rates and subsequent workload as well as increased complexity [6, 7]. This rise in general practice healthcare utilisation has not been matched by a corresponding growth in the workforce capacity [8], and research findings with GP participants warned about both patient and doctor safety in this context [9]. The burden of workload and symptoms of burnout among GPs have been reported in Ireland [10] and other European countries [11], with one study reporting an association between GP burnout and the prevalence of multi-morbidity in the practice [12].

Multi-morbidity, defined as having at least two chronic diseases [13], is present in over one quarter of adults registered at general practices (and is present in a much higher proportion of those attending general practice) and is associated with higher attendance at general practice and hospital [14]. Multi-morbidity is strongly associated with increasing age and with populations in western societies becoming older, the prevalence of multi-morbidity is rising and this trend can be expected to continue [15, 16]. Similarly, polypharmacy, which is defined as taking five or more regular medications [17], is associated with chronic illness and multi-morbidity [18], older age [19] and increased health care utilisation [20]. While specialists provide disease specific management and prescribe system specific medications, GPs co-ordinate the care, medication prescribing and follow up of their patients. In many systems, such as Ireland and Britain, GPs act as gatekeepers to the hospital system, assessing and managing untriaged and undifferentiated presentations and initiating referrals to the hospital system. Much of the work involved in managing polypharmacy and multi-morbidity takes place in general practice [8]. It is very important for healthcare planning that this work is reported in a clear way that illustrates the amount and type of activity that is conducted as well as the rate of referral to hospital doctors.

While several largescale studies have reported on health care utilization in general practice and referral rates to hospital doctors, there has been criticism of the methods used. Researchers have reported that dependency on certain coding systems may have missed out on un-coded illness and that the inclusion criteria for chronic illness was, in some instances, too narrow in its scope and thus failed to include all chronic conditions [14]. Therefore, the manual searching of patient records to investigate practice network or national databases is considered to be preferable as they overcome the limitations of coding and facilitate a deeper insight into the

notes to determine patient factors and how they impacted management [21]. Conversely, this method may not account for attendances to emergency departments that did not involve a GP referral, but many of these visits may be recorded in the notes when the relevant hospital discharge letter is filed. Furthermore, the reporting in other studies has failed to distinguish between nurse and GP consultations as well as in-practice consultations and home visits [22], the latter being an important but time-consuming task for GPs [23]. The role of the practice nurse does not involve making referrals to hospitals [24], and it is, therefore, important to capture data on who exactly in the practice the patient attended when calculating referral rates. Precise data on the work taking place in general practice is important to know to understand the role of general practice in the health service. This study aimed to provide comprehensive data on healthcare utilisation in general practice and referrals to hospital. Specific objectives were: to determine the impact of age, chronic illness and number of regularly prescribed medications on consultation rates with the GP, practice nurse and on home visits and referral rates to hospital.

## Materials and methods

### Study population

This study was a retrospective analysis of consultations and was granted full ethical approval by the Irish College of General Practitioners Research Ethics Committee (ICGP, 09/05/2015). As per the ethics application, all data were fully anonymised before leaving the practice and before being accessed for analysis. Individual informed consent was not deemed necessary by the ethics committee. The study took place over a two-year period in general practices associated with the University of Limerick Education and Research Network for General Practice (ULEARN-GP) [25]. All 72 practices in the network with a student on placement in 2015/16 were invited to participate. At the time of the study, the network covered three of Ireland's four health regions, and was representative of the national demographic in terms of size, personnel, urban-rural mix, age and socio-economic profiles [25, 26]. Participating practices were asked to use practice software to extract a random sample of 100 patients aged 50 years and older that had attended the practice at least once in the previous two years. This study was part of a larger investigation of processes of care and communication between general practice and hospitals [27]. Senior medical students on placement in the practices in conjunction with their GP supervisors were trained by a faculty team on how to select the sample, and how to search their medical records for the relevant data.

### Data collection

After appropriate training, data were extracted, anonymised, coded and entered onto a Microsoft Excel document by each student for each patient selected in the sample. Students were taught by faculty how to use practice software to extract a list of patients over 50 years of age who had attended the practice in the previous two years. They were also shown how to use a randomisation function on the software to extract 100 patients for inclusion in the study. For each patient record, entries for a two-year period extending from 1st September 2013 to 31st August 2015 were analysed. Medical records were searched for the presence of chronic illness through disease coding, free text entry or documentation in expert reports from hospital or consultation records. The number of chronic illnesses and number of regularly prescribed medications was recorded. Chronic illness was defined as a long-term medical condition that cannot be completely cured by medicines; a list of chronic illnesses compiled by the clinicians on the research team and based on a list utilised by a national longitudinal study [28] was provided for the students (S1 Table). Demographic data was collected on each patient, including

gender, age, and eligibility for a General Medical Services (GMS) card. This card is given on a means tested basis to individuals and families with lower incomes and at the time of the study approximately 43% of the population were eligible for a GMS card [29]. The income thresholds are higher for those aged over 70.

Health care utilisation data recorded included: number of visits to the GP, number of practice nurse visits, number of home visits, number of referrals to hospital doctors, including Emergency Department, specialist outpatients and injury assessment units. Referrals for radiology and other diagnostic procedures that did not involve a consultation with a hospital doctor were recorded separately.

## Data analysis

Data were coded onto Microsoft Excel spreadsheets in each practice and only completely anonymised data was submitted by the practices to the research team for analysis. Only data that had complete demographic details were included in the analysis. Data was described using counts and percentages for categorical variables; mean (standard deviation) for normally distributed numeric variables; and median (interquartile range) for skewed distributions. Pearson's correlation coefficient was used to measure the strength of the association between numeric variables. Attendance and referral rates were presented by calculating the rate per person-year with 95% confidence intervals for each demographic variable separately. The ratio of attendance to the GP and hospital referral was calculated. A chi-square test was used to test the association between categorical variables. A significance level of 5% was used for all tests. The strength of the association was measured using Cramer's V with a value of $< 0.2$ considered weak, 0.2 to 0.6 considered moderate and $>0.6$ considered a strong association [30]. Statistical analysis was conducted using IBM SPSS version 26.

## Results

### Practice and patient characteristics and referral rates

Sixty-eight (94%) of the 72 practices that were invited agreed to participate in the study, yielding a total of 6800 patients' records to be evaluated. Of these, 197 records (3%) were excluded as insufficient demographic data was recorded. Data over the two-year study period was available for analysis in 6603 records (13,206 person-years). Over half the patients (57%) were eligible for a medical card and eligibility increased with age (88% of those aged 70 and over). Approximately half (52%) were female, and the median age was 63 (IQR 56–72) years.

All of the practices were mixed public-private, were computerised and had a practice nurse. Table 1 compares the profile of participating practices to the national profile in 2015 [26]. Most of the study practices (93%) had a co-operative system of out-of-hours cover, similar to the national profile (92%). Participating practices had higher percentages involved in postgraduate GP training (43% v 22%) and rural location (37% v 21%). In terms of practice size, 68% had between 500 and 1999 patients; 16% were single-handed practices, 31% had two GPs, 24% had three GPs and 29% had four or more GPs.

The median number of chronic conditions was 1 (IQR 0–2) and the median number of medications was 3 (IQR 1–7). Age was positively correlated with the number of chronic diseases (r = 0.37, p<0.001) and the number of medications (r = 0.46, p<0.001). The number of chronic illnesses and the number of medications were strongly positively correlated (r = 0.67, p<0.001). The prevalence of multi-morbidity was 38% and the prevalence of polypharmacy was 39%. In the previous two years, 3310 (50%) had been referred to hospital at least once. Likelihood of referral increased with age eligibility for a GMS card, number of chronic diseases and number of prescribed medications (Table 2). The strongest associations with being

**Table 1. Comparison of 2015 national profile to practice profile.**

|  | National, 2015 [26] | ULEARN-GP participating practices, 2015 |
|---|---|---|
| Number of practices | 462 | 68 |
| Practice type |  |  |
| Mixed GMS and private | 89% | 100% |
| Private only | 11% | 0% |
| GMS list size |  |  |
| <500 | 18% | 16% |
| 500–1999 | 75% | 68% |
| >2000 | 7% | 16% |
| Practice location |  |  |
| Rural | 21% | 37% |
| Urban | 42% | 28% |
| Mixed | 37% | 35% |
| Premises |  |  |
| Purpose-built | 54% | 35% |
| Adapted/ other | 46% | 65% |
| Practice operation |  |  |
| Computerisation | 94% | 100% |
| Out of hours |  |  |
| Internal rota | 1% | 0 |
| External rota | 6% | 8% |
| Co-operative | 93% | 92% |
| Practice staff |  |  |
| Single-handed GP | 18% | 16% |
| Practice nurse | 82% | 100% |
| Education |  |  |
| Involved in post-graduate training | 22% | 43% |

referred were with number of chronic diseases and number of prescribed medications (Cramer's V > 0.2, Table 2) and the highest proportion of those with any referral was for patients with five or more prescribed medications (66%).

## Consultation data and referral rates

A total of 89,667 practice consultations were recorded over two years for the 6603 patients. Of these, 1253 (1.4%) were home visits by the GP and 23,110 (26%) were attendances to the practice nurse. There were 65,304 attendances to the GP in 13,206 person-years which gives a rate of 4.94 per person-year (95% confidence interval 4.91 to 4.98) i.e. a patient aged 50 years or over, on average attended the GP five times per year. There were 7,859 hospital referrals, giving a referral rate of 0.60 per person-year (95% confidence interval 0.58 to 0.61) i.e. a patient over 50 on average was referred less than once a year to a hospital. The consultation rate of 4.94 to the GP is over eight times the referral rate of 0.60 in these patients aged 50 or over. Table 3 summarises number of attendances and referrals by age group, gender, GMS eligibility, number of chronic conditions and prescribed medications. Females attended the GP more than males, had over double the rate of home visits and had a higher referral rate. The ratio of GP attendance to referral rate was, however, similar for both males and females. Patients eligible for a GMS card had higher rates of GP and nurse attendance, home visits and hospital referral rates which may reflect their older age profile.

**Table 2. Characteristics of patients by referral status.**

| Characteristics | No referral to hospital n = 3310 (50%) | At least one referral to hospital n = 3293 (50%) | p-value (Cramer's V) |
|---|---|---|---|
| **Gender** | | | 0.02 (0.03) |
| Female | 1659 (48.7%) | 1746 (51.3%) | |
| Male | 1603 (51.7%) | 1495 (48.3%) | |
| **Age group** | | | <0.001 (0.13) |
| 50–59 | 1428 (57.0%) | 1078 (32.7%) | |
| 60–69 | 1006 (50.5%) | 985 (49.5%) | |
| 70–79 | 570 (41.9%) | 791 (58.1%) | |
| ≥ 80 | 306 (41.1%) | 439 (58.9%) | |
| **GMS eligibility** | | ' | <0.001 (0.16) |
| Eligible | 1645 (43.4%) | 2146 (56.6%) | |
| Non-eligible | 1665 (59.3%) | 1145 (40.7%) | |
| **Number of chronic illnesses** | | | <0.001 (0.23) |
| None | 1555 (63.8%) | 882 (36.2%) | |
| One | 843 (50.1%) | 841 (49.9%) | |
| Two or more | 912 (36.7%) | 1571 (63.3% | |
| **Number of regularly prescribed medications** | | | <0.001 (0.29) |
| None | 1087 (72.7%) | 408 (27.3%) | |
| 1–4 | 1347 (52.6%) | 1216 (47.4%) | |
| Five or more | 876 (34.4%) | 1669 (65.6%) | |

**Table 3. Attendances and referrals by gender, GMS eligibility, age group, number of chronic illnesses and prescribed medications.**

| | GP attendances per person-year (95% CI) | Home visits per person-year (95% CI) | Nurse attendances per person-year (95% CI) | Hospital referrals per person-year (95% CI) | Ratio of GP attendances/ referral |
|---|---|---|---|---|---|
| **Gender** | | | | | |
| Male (n = 3098) | 4.58 (4.53, 4.64) | 0.06 (0.05, 0.07) | 1.78 (1.75, 1.82) | 0.53 (0.51, 0.55) | 8.6 |
| Female (n = 3405) | 5.30 (5.24, 5.35) | 0.13 (0.12, 0.14) | 1.72 (1.69, 1.75) | 0.64 (0.62, 0.66) | 8.3 |
| **GMS eligibility** | | | | | |
| Eligible (n = 3791) | 6.57 (6.52, 6.63) | 0.15 (0.14, 0.15) | 2.45 (2.42, 2.49) | 0.74 (0.72, 0.76) | 8.9 |
| Non-eligible (n = 2810) | 2.74 (2.70, 2.79) | 0.03 (0.02, 0.03) | 0.81 (0.78, 0.83) | 0.40 (0.38, 0.41) | 6.9 |
| **Age group** | | | | | |
| 50–59 (n = 2506) | 3.59 (3.54, 3.64) | 0.01 (0.01, 0.02) | 1.00 (0.97, 1.03) | 0.45 (0.43, 0.47) | 8.0 |
| 60–69 (n = 1991) | 4.55 (4.48, 4.61) | 0.04 (0.03, 0.04) | 1.56 (1.52, 1.60) | 0.59 (0.57, 0.62) | 7.7 |
| 70–79 (n = 1361) | 6.44 (6.34, 6.54) | 0.11 (0.10, 0.12) | 2.49 (2.44, 2.55) | 0.74 (0.71, 0.78) | 8.7 |
| ≥ 80 (n = 745) | 7.85 (7.70, 7.99) | 0.50 (0.46, 0.53) | 3.41 (3.32, 3.50) | 0.82 (0.78, 0.87) | 9.6 |
| **Number of chronic illnesses** | | | | | |
| None (n = 2437) | 2.70 (2.65, 2.75) | 0.03 (0.02, 0.03) | 0.73 (0.71, 0.75) | 0.37 (0.35, 0.38) | 7.3 |
| One (n = 1684) | 4.52 (4.45, 4.59) | 0.07 (0.06, 0.08) | 1.38 (1.34, 1.42) | 0.54 (0.52, 0.57) | 8.4 |
| Two or more (n = 2482) | 7.44 (7.36, 7.52) | 0.18 (0.17, 0.19) | 3.00 (2.95, 3.05) | 0.86 (0.83, 0.88) | 8.7 |
| **Number of prescribed medications** | | | | | |
| None (n = 1495) | 1.74 (1.69, 1.78) | 0.01 (0.01, 0.02) | 0.42 (0.40, 0.45) | 0.21 (0.20, 0.23) | 8.3 |
| 1–4 (n = 2563) | 4.00 (3.95, 4.06) | 0.03 (0.03, 0.04) | 1.28 (1.25, 1.32) | 0.49 (0.47, 0.51) | 8.2 |
| Five or more (n = 2545) | 7.77 (7.70, 7.85) | 0.21 (0.20, 0.22) | 3.00 (2.95, 3.05) | 0.93 (0.90, 0.95) | 8.4 |

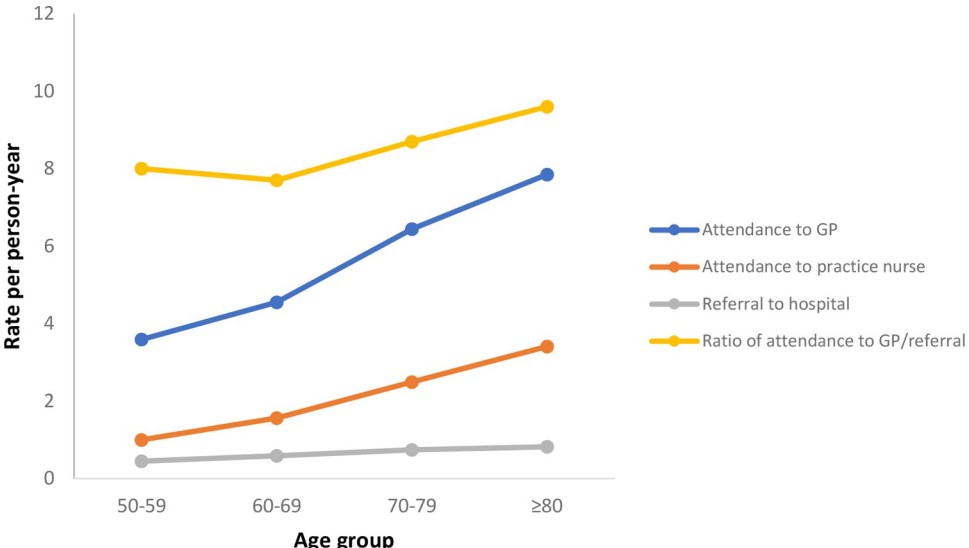

**Fig 1. Attendances and referrals by age group.**

Table 3 and Fig 1 illustrate that with each 10-year increase in age, the rates of attendance to the GP and practice nurse increase. The ratio of GP attendances to hospital referrals was 8.0 for patients aged 50–59 and 9.6 for patients aged 80 years and over, indicating slightly more GP attendances per referral as age increases.

Fig 2 categorises the number chronic illnesses from none through to six or more. The attendance rates to the nurse and GP rise with each additional chronic illness but the ratio of GP attendance to referral rate remains relatively stable, e.g. 8.4 for patients with one chronic illness and 8.7 for patients with six or more chronic illnesses.

The number of regularly prescribed medications is associated with increased GP and nurse attendances and Fig 3 illustrates a sharp increase in both attendance rates with five or more

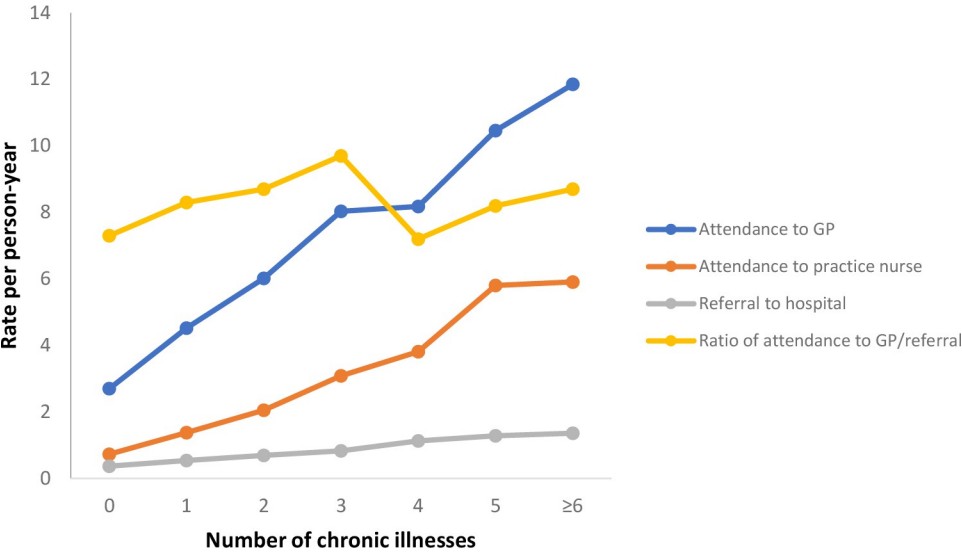

**Fig 2. Attendances and referrals by number of chronic illnesses.**

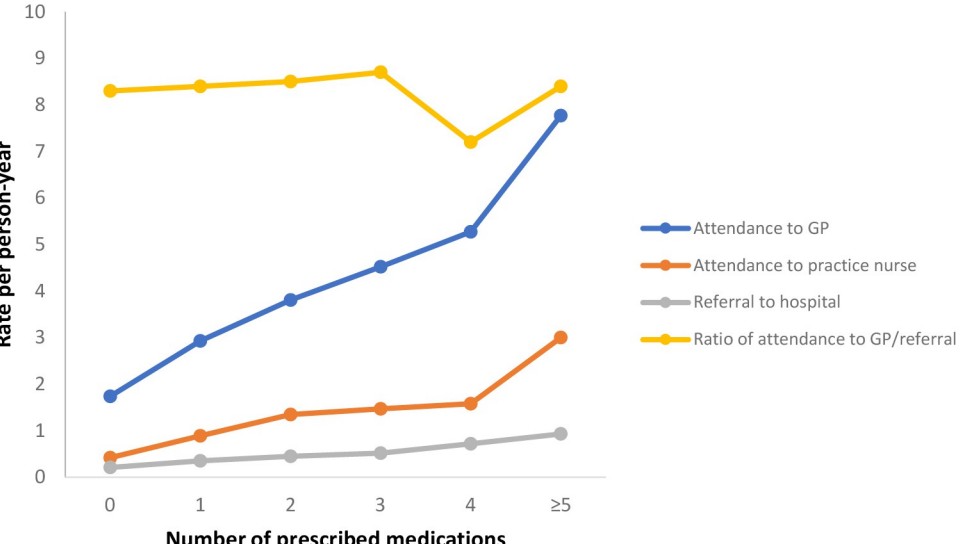

**Fig 3. Attendances and referrals by number of prescribed medications.**

medications. The ratio of GP attendance to hospital referral remains relatively stable at 8.3 in a patient prescribed no medication and 8.4 for a patient prescribed five or more medications.

## Discussion

### Summary of main findings

This paper has captured one of the unique capabilities of general practice–its ability to deal with increasing age and medical complexity of patients without relying on more referrals to hospital specialists. The study was set in 68 general practices across the Republic of Ireland and involved an analysis of the medical records of 6800 patients aged 50 years and over, yielding data on a total of 89,667 practice consultations. It included over 20,000 practice nurse visits and over 1,000 home visits by GPs. The rates of GP attendance and referral to hospital were reported per person-year and were 5 and 0.6 respectively. This meant that for one referral to the hospital there were eight attendances to the GP, excluding consultations that took place in patients' homes and practice nurse consultations.

Approximately one half of the study population was referred to hospital doctors at least once in the two-year time period. The strongest associations with any referral were with number of chronic diseases and number of prescribed medications. Increasing age, number of chronic illnesses and number of medications was associated with increased attendances to the GP and practice nurse as well as the number of home visits. However, the ratio of attendance to the GP and referral to hospital remained relatively stable between 6.9 and 9.6. These figures indicate that much of the management of older patients, patients with considerable multimorbidity or polypharmacy are managed in general practice and that much of the workload associated with additional morbidity and medication is conducted by the general practice team.

### Comparison to the literature

The GP consultation rate of five per person per year was very similar to that reported in Britain as were the higher attendance rates with increasing age and among those with higher numbers of chronic illnesses [2]. The relatively stable ratio of attendances to hospital referrals reported

in our study is similar to previous research on the impact multimorbidity on healthcare utilisation conducted in Ireland [31]. As these studies reported all primary care consultations, including practice nurse and GP visits, as a single entity we cannot make direct comparisons on attendances other than the shared conclusion that as morbidity increased so too did attendance to general practice in each study [22, 31].

An international study across 16 countries found that as the number of chronic illnesses increased so too did the use of primary and secondary health care utilisation [32]. The methodology used a self-report survey and so cannot be directly compared to our study. It also took place in countries with different type of health care system structures with varying degrees of gatekeeping, meaning that GPs would not necessarily decide on the secondary care pathway for patients. Similar to our study, the population was aged 50 years and older, most likely chosen because of their increased susceptibility to chronic disease [33]. Analysis of a large clinical practice research database in England reported higher health care utilisation in general practice, higher number of prescription medications and higher hospitalisations with multi-morbidity [14]. To our knowledge, no study to date has reported the association between polypharmacy, health care utilisation in primary care, and referral to hospital.

The study findings suggest that much of the multi-morbidity is managed in general practice. GPs provide person centred care, through highly developed doctor-patient relationships and continuity of care and GPs are known to provide individually tailored management plans whereby they collaborate with patients to agree on self-management and pharmacological options [34, 35]. However, serious threats to this model exist [36], including the capacity of general practice to continue to absorb the workload involved. These consultations take more time than single complaint presentations [37] and administrative time for medication reviews [34] and these factors must be considered in funding models for general practice. Managing polypharmacy in the context of providing holistic care to each individual when the available guidelines are focussed on single diseases is a challenge for GPs [1, 37]. Most guidelines recommend appropriate management for individual diseases but do not reflect how the presence of other chronic illnesses might affect prescribing decisions. Guidelines that are patient-centred and that are based on chronic disease clusters would be more useful for GPs [38]. It is important also to recognise the value of clinical judgement [39], as the social context, disease presentation and pattern will be different for each patient.

## Strengths and limitations

The study involved detailed analysis of patient records, extracting data from letters from hospital consultants, radiology reports and free text entries, in addition to disease coding, thereby providing a detailed picture of the chronic illness status of the study population. This approach of using multiple sources has been considered advantageous rather than relying on adherence to coding by the GPs only. However, as stated, the approach may miss self-presentations to the hospital, especially to emergency departments, and such presentations were not included in this study. The high participation rate (94%) of a representative sample of GP practices contributes to external validity. The breakdown of consultations in general practice into attendance at the GP, attendance at the nurse and home visits gives a detailed picture of the workflow in this setting. Out of hours consultations were not included as there was no consistency across practices on how this information was recorded and stored. Further, information was recorded on reason for referral so that patients referred for radiology and other diagnostics could be differentiated from those referred for assessment by a hospital consultant or team.

Our focus was on older adults with at least one attendance to the practice in the previous two years. We aimed to include active patients in the practice and minimise those who had potentially moved away or changed practices with no recent record of attendance. Our approach, however, may overestimate healthcare utilisation by excluding those with no recent record of attending. An additional limitation was that each practice had a separate data collector. Even though these were all trained for consistency, it is possible this could have led to inter-rater variability in terms of data collection process. Irish general practice software systems are not generally used by other healthcare professionals working in primary care such as physiotherapists, dieticians and healthcare assistants; this study involves GPs and practice nurses only. Further, this study used a definition of multi-morbidity of two or more chronic conditions; those chronic conditions were based on a pre-defined list that was provided for the students and set by the clinicians on the research team, and conditions not on the list were not included in the dataset. Finally, during data collection the number of chronic conditions and medications were recorded for each case but not the name of the chronic condition or medication, and, consequently, the analysis cannot identify which chronic conditions or medications were associated with attendance.

## Implications for future research and practice

The authors believe that future research, analysing patient records, should investigate the impact of specific clusters of chronic illness, especially mental illness, on health care utilisation patterns in general practice. Training for GPs focussed on coding chronic illness as well as developments in software to make coding easier or invisible, would improve the accuracy and reach of health services research in general practice. The study was conducted prior to the Covid 19 pandemic and, no doubt, telemedicine, including video consults, telephone consults and consults using other technology to send audio-visual files, will be much more frequent into the future. GPs with the support of software developers should be able to record this workflow in a way that is user-friendly, minimally time consuming and easily extractable for future research. Finally, the workload conducted by reception staff, including phone-calls, triage, advice and information giving must be captured and presented in order to more completely illustrate the entire burden of work complex multimorbidity presents for general practice.

## Conclusion

This large-scale study of individual patient records has provided a detailed and precise picture of the quantity and type of work taking place in general practice. We have shown that approximately one half of patients aged 50 years and over who attended the GP in the previous two years were referred to hospital. As age, morbidity and medication numbers rise, so too do all types of consultations in general practice but the ratio of GP attendance to referral rate remains relatively stable, indicating that GPs are managing these patients without increased referral rate per consultation. The implications of this finding are extremely important, as it demonstrates that general practice is bearing most of the burden of increased morbidity and complexity, thereby absorbing excess workload and saving hospital outpatients appointments, emergency department presentations and hospital admissions. Consequently, there are benefits for healthcare economics as well as for the lives of patients who can be managed in the community. General practice must be supported to develop its capacity to provide person centred, individually-tailored care to an ageing population with rising rates of multi-morbidity and polypharmacy.

## Supporting information

**S1 Table. List of chronic conditions.**
(PDF)

## Acknowledgments

The authors would like to acknowledge the contribution of the general practitioner tutors and their practice staff as well as the students on placement who were involved in data collection.

## Author Contributions

**Conceptualization:** Andrew O'Regan, Walter Cullen, Ailish Hannigan.

**Data curation:** Andrew O'Regan, Jane O'Doherty.

**Formal analysis:** Andrew O'Regan, Ailish Hannigan.

**Methodology:** Andrew O'Regan, Ray O'Connor, Walter Cullen, Vikram Niranjan, Liam Glynn, Ailish Hannigan.

**Project administration:** Andrew O'Regan, Jane O'Doherty, Vikram Niranjan.

**Supervision:** Liam Glynn, Ailish Hannigan.

**Validation:** Ray O'Connor, Walter Cullen, Liam Glynn, Ailish Hannigan.

**Writing – original draft:** Andrew O'Regan, Jane O'Doherty, Ailish Hannigan.

**Writing – review & editing:** Andrew O'Regan, Jane O'Doherty, Ray O'Connor, Walter Cullen, Vikram Niranjan, Liam Glynn, Ailish Hannigan.

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
