## [Decision Letter · Decision Letter 0]

25 Jun 2021

PONE-D-21-15158

How do multi-morbidity and polypharmacy affect general practice attendance and referral rates? A retrospective analysis of consultations.

PLOS ONE

Dear Dr. O'Regan,

Thank you for submitting your manuscript to PLOS ONE. After careful consideration, we feel that it has merit but does not fully meet PLOS ONE’s publication criteria as it currently stands. Therefore, we invite you to submit a revised version of the manuscript that addresses the points raised during the review process.

We look forward to receiving your revised manuscript.

Kind regards,

Vijayaprakash Suppiah, PhD

Academic Editor

PLOS ONE

Journal Requirements:

2.  Please provide additional details regarding participant consent. In the ethics statement in the Methods and online submission information, please ensure that you have specified (a) whether consent was informed and (b) what type you obtained (for instance, written or verbal, and if verbal, how it was documented and witnessed). If your study included minors, state whether you obtained consent from parents or guardians. If the need for consent was waived by the ethics committee, please include this information.

Reviewers' comments:

Reviewer's Responses to Questions

**Comments to the Author**

1. Is the manuscript technically sound, and do the data support the conclusions?

Reviewer #1: Yes

Reviewer #2: Yes

2. Has the statistical analysis been performed appropriately and rigorously? 

Reviewer #1: Yes

Reviewer #2: Yes

3. Have the authors made all data underlying the findings in their manuscript fully available?

Reviewer #1: No

Reviewer #2: Yes

4. Is the manuscript presented in an intelligible fashion and written in standard English?

Reviewer #1: Yes

Reviewer #2: Yes

5. Review Comments to the Author

Reviewer #1: Dear Editor,

thank you for asking me to review this paper submission. Comments:

1) Abstract: Methods not clearly described. How practices were in the Research Network (72 practices were invited to participate but were there lots more practices who were not included? If so, how were the 72 selected?). If 68 practices selected, each contributing 100 patients, why isnt the sample 6800 patients? Were findings 'adjusted' or 'unadjusted', e.g. for age, gender, deprivation; it would be helpful to know in the Abstract if this is a univariate or multivariate analysis.

2) Introduction: Reference 5 is cited as evidence of 'the recent funding strategy' for primary care. However, it's worth clarifying that much of this funding in England will be allocated to Primary Care Networks (practices working collaboratively within a locality with a population health agenda) rather than being allocated to individual practices in the traditional way.

3) Introduction: Reference 14 is misquoted. The authors state: 'Multi-morbidity, defined as having at least two chronic diseases (13), is present in over one quarter of adults attending general practice...(14)'. In fact, multimorbidity is present in over a quarter of those REGISTERED at general practices, but account for a much higher proportion of those ATTENDING the practice.

4) Introduction: the authors state: 'Therefore, the manual searching of patient records to investigate practice network or national databases is considered to be preferable as they overcome the limitations of coding...'. This provides a strong justification for this study. Studies of coded data are likely to miss cases of multimorbidity (MM). However, the converse should be acknowledged - that a study purely of primary care data will miss emergency presentations to secondary care, which bypass primary care (although these should be recorded retrospectively in primary care case-notes).

5) Methods: as per the Abstract, it's not clear if the Network of GP practices consisted of more than 72 practices. Nor why <100 cases were recruited in each practice

6) Methods: the authors used Cramer’s V to measure strength of association. This is not a commonly used test. How does it differ from Pearson's r value? My reading is that this study has used univariate parametric correlation coefficients, but I may be mistaken. It is important that the readers know how the data was analysed (univariate presumably, parametric or non-parametric associations?)

7) Results: we need a clear definition of MM. Which MM's were included and which were not? Without that list, it's not possible to make sense of the finding: 'The prevalence of multi-morbidity was 38%...'

8) Results, Figure 1. The authors state: 'Fig 1 illustrates that with each 10-year increase in age, the rates of attendance to the GP and practice nurse increase.' But they haven't demonstrated this, at least not without further Stats analysis. What if the mean attendance rate confidence intervals overlapped for each 10-year attendance rate value? Without CI's or a test of difference, the authors cannot conclude that there is a difference in these rates. The same applies to the data shown in Figures 2&3. This is an essential revision required before publication.

9) Discussion: The authors do have a key finding that is of importance: 'However, the ratio of attendance to the GP and referral to hospital remained relatively stable between 6.9 and 9.6. These figures indicate that much of the management

of older patients, patients with considerable multimorbidity or polypharmacy are managed in general practice.' This is important and needs emphasis.

10) Discussion, Strengths and Limitations: the authors state: ' This approach of using multiple sources has been considered advantageous rather than relying on adherence to coding by the GPs'. I suspect the authors don't have the all-important data which would provide a strong justification for their approach. Namely, how much of the data they found were buried away in un-coded data? That would justify manual data searching as in this study, and as opposed to electronic data searching of coded (no freetext) anonymised databases such as CPRD.

11) Discussion, Strengths and Limitations: there are many limitations which are not mentioned. A key limitation is how MM was classified (what was and what was not an MM). Why weren't Health Care Assistant consultations included or other healthcare providers within primary care (such as dietitians, midwives, physios etc, if present)? Also, as stated above, what about patients accessing hospital care by bypassing their GP, e.g. emergency admissions - was this data retrospectively captured through inclusion of hospital discharge summaries (this isnt clear in the Methods).

only'

Reviewer #2: The study objectively shows what many GPs feel on a daily routine: the workload is increasing, the patient cases are becoming more and more complex. Thus, GPs need new tools to be able to treat patients as efficiently and professionally as possible. In my view, the most important result of their study is the constant rate of referrals to hospital. The protective effect for the clinics should not be underestimated. The results of the study are comprehensible and answer the research question. Essentially, I still miss a more detailed description of the data collected:

What were the characteristics of the participating GP practices? Number of doctors, age, number of patients, etc.

Is there information on the most common diseases or medication groups that led to the definition of multimorbidity and polypharmacy?

In which specialist departments were the patients referred?

Are there diseases that led to particularly frequent referrals - where did outpatient care in GP practices reach its medical limits?

Perhaps additional tables could be used to describe the cohort studied even better and thus make it more comparable for other international data!?.

6. PLOS authors have the option to publish the peer review history of their article (what does this mean?). If published, this will include your full peer review and any attached files.

Reviewer #1: **Yes: **Mark Ashworth

Reviewer #2: **Yes: **Markus Bleckwenn

---

## [Author Response · Author response to Decision Letter 0]

19 Aug 2021

19.08.2021

Dear editor and reviewers,

We are very grateful for your time in reading and reviewing this manuscript. We believe that the suggestions received are valuable and we have responded to each of them in detail below. For each response, the corresponding alteration in the manuscript in quoted, where appropriate. 

We would like to point out that this was a retrospective analysis of patient records. Each one of the 72 practices affiliated with the Medical School that had a medical student on placement was invited to participate and 68 of them accepted. Of the 6800 records a small percentage were discounted for not having adequate demographic data. 

We had full ethical approval for the study. No anonymised data left the practice at any stage and no approval exists for publishing the dataset.

Statistical queries, including confidence intervals and use of Cramer’s V test, have been addressed.

Finally, the research team has a prior publication from this dataset, which has been referenced, and which gives more detail on the destination of the referrals, including hospital specialty.

We hope that the comments have been addressed to the satisfaction of the reviewers.

Best wishes,

Dr Andrew O’Regan, GP and senior lecturer in general practice, on behalf of the research team

Journal Requirements:

2. Please provide additional details regarding participant consent. In the ethics statement in the Methods and online submission information, please ensure that you have specified (a) whether consent was informed and (b) what type you obtained (for instance, written or verbal, and if verbal, how it was documented and witnessed). If your study included minors, state whether you obtained consent from parents or guardians. If the need for consent was waived by the ethics committee, please include this information.

Response of the authors: In the subsection ‘Study Population’ of ‘Materials and Methods’, the authors have clarified as follows:

“This study was a retrospective analysis of consultations and was granted full ethical approval by the Irish College of General Practitioners Research Ethics Committee (ICGP, 09/05/2015). As per the ethics application, all data were fully anonymised before leaving the practice and before being accessed for analysis. Individual informed consent was not deemed necessary by the ethics committee.”

Response of the authors: sharing of this dataset is restricted by the Irish College of General Practitioners research ethics committee. Access to the dataset is available upon reasonable request to the Irish College of General Practitioners Research Ethics Committee (research@icgp.ie).

 Response of the authors: same as point a) above.

Reviewers' comments:

Reviewer's Responses to Questions

Comments to the Author

1. Is the manuscript technically sound, and do the data support the conclusions?

Reviewer #1: Yes

Reviewer #2: Yes

2. Has the statistical analysis been performed appropriately and rigorously?

Reviewer #1: Yes

Reviewer #2: Yes

3. Have the authors made all data underlying the findings in their manuscript fully available?

Reviewer #1: No

Reviewer #2: Yes

4. Is the manuscript presented in an intelligible fashion and written in standard English?

Reviewer #1: Yes

Reviewer #2: Yes

5. Review Comments to the Author

Reviewer #1: Dear Editor,

thank you for asking me to review this paper submission. Comments:

1) Abstract: Methods not clearly described. How practices were in the Research Network (72 practices were invited to participate but were there lots more practices who were not included? If so, how were the 72 selected?). If 68 practices selected, each contributing 100 patients, why isnt the sample 6800 patients? Were findings 'adjusted' or 'unadjusted', e.g. for age, gender, deprivation; it would be helpful to know in the Abstract if this is a univariate or multivariate analysis.

Response of the authors: We have clarified the practice research network query in the abstract and in the main manuscript as follows:

Abstract 

“This was a retrospective study of general practices in a university-affiliated education and research network, consisting of 72 practices.”

Main manuscript

“This study was a retrospective analysis of consultations that took place over a two-year period in general practices associated with the University of Limerick Education and Research Network for General Practice (ULEARN-GP) (25). All 72 practices in the network with a student on placement in 2015/16 were invited to participate.”

We have clarified that 100 patients were selected at random from each practice, resulting in a sample of 6800 patients. A few practices did not provide basic demographic data (age, GMS status) on all 100 patients and these patients were excluded (n =197, 3% of sample). With the word count restriction in the Abstract we have clarified this point in the methods section.

Bivariate associations are given in Table 2 and referral rates are provided by each demographic variable separately (age, gender, etc.). We have clarified this in the Abstract and Methods.

2) Introduction: Reference 5 is cited as evidence of 'the recent funding strategy' for primary care. However, it's worth clarifying that much of this funding in England will be allocated to Primary Care Networks (practices working collaboratively within a locality with a population health agenda) rather than being allocated to individual practices in the traditional way.

Response of the authors: the authors thank the reviewer for pointing out this important distinction. We have clarified in the text as follows:

“In England, a recent funding strategy will support Primary Care Networks, whereby practices work collaboratively within a locality with a population health agenda, and it remains to be seen if this can successfully alleviate the pressure on this key component of the health service.” 

3) Introduction: Reference 14 is misquoted. The authors state: 'Multi-morbidity, defined as having at least two chronic diseases (13), is present in over one quarter of adults attending general practice...(14)'. In fact, multimorbidity is present in over a quarter of those REGISTERED at general practices, but account for a much higher proportion of those ATTENDING the practice.

Response of the authors: the authors again thank the reviewer for pointing out this important distinction. We have clarified in the text as follows:

“Multi-morbidity, defined as having at least two chronic diseases (13), is present in over one quarter of adults registered at general practices (and is present in a much higher proportion of those attending general practice) and is associated with higher attendance at general practice and hospital (14).”

4) Introduction: the authors state: 'Therefore, the manual searching of patient records to investigate practice network or national databases is considered to be preferable as they overcome the limitations of coding...'. This provides a strong justification for this study. Studies of coded data are likely to miss cases of multimorbidity (MM). However, the converse should be acknowledged - that a study purely of primary care data will miss emergency presentations to secondary care, which bypass primary care (although these should be recorded retrospectively in primary care case-notes).

Response of the authors: we have acknowledged this point in the introduction:

“Conversely, this method may not account for attendances to emergency departments that did not involve a GP referral, but many of these visits may be recorded in the notes when the relevant hospital discharge letter is filed.”

5) Methods: as per the Abstract, it's not clear if the Network of GP practices consisted of more than 72 practices. Nor why <100 cases were recruited in each practice

Response of the authors: we have clarified that all 72 affiliated practices with a student on clinical placement in 2015/16 were invited. We only included cases that had complete demographic data. We have clarified this point in the methods section and again in the results section.

Methods: “Only data that had complete demographic details were included in the analysis.”

Results: “197 records (3%) were excluded as insufficient demographic data was recorded.”

6) Methods: the authors used Cramer’s V to measure strength of association. This is not a commonly used test. How does it differ from Pearson's r value? My reading is that this study has used univariate parametric correlation coefficients, but I may be mistaken. It is important that the readers know how the data was analysed (univariate presumably, parametric or non-parametric associations?)

Cramer’s V is analagous to Pearson’s r but used for categorical variables (all variables in Table 1) instead of the continuous variables needed for Pearson’s r. Cramer’s V is a measure of the strength of the association between two categorical variables, independent of sample size, and values range from 0 to 1. We have now provided a reference for it (Cohen, J. 1988. Statistical Power Analysis for the Behavioral Sciences, 2nd Edition. Routledge).

7) Results: we need a clear definition of MM. Which MM's were included and which were not? Without that list, it's not possible to make sense of the finding: 'The prevalence of multi-morbidity was 38%...'

Response of the authors: we defined multi-morbidity in the introduction as “having at least two chronic illnesses.” [Reference: Le Reste JY, Nabbe P, Manceau B, Lygidakis C, Doerr C, Lingner H, et al.]

Also added to the Methods section: “A list of chronic illnesses, which were defined as a long-term medical condition that cannot be completely cured by medicines, was provided for the students (see supplementary table 1)”. This list is provided now as a supplementary file.

8) Results, Figure 1. The authors state: 'Fig 1 illustrates that with each 10-year increase in age, the rates of attendance to the GP and practice nurse increase.' But they haven't demonstrated this, at least not without further Stats analysis. What if the mean attendance rate confidence intervals overlapped for each 10-year attendance rate value? Without CI's or a test of difference, the authors cannot conclude that there is a difference in these rates. The same applies to the data shown in Figures 2&3. This is an essential revision required before publication.

Response of the authors: We thank the reviewer for this important point. We have now added confidence intervals to all the estimates in Table 2. The confidence intervals are narrow and not overlapping and support the conclusions above.

9) Discussion: The authors do have a key finding that is of importance: 'However, the ratio of attendance to the GP and referral to hospital remained relatively stable between 6.9 and 9.6. These figures indicate that much of the management

of older patients, patients with considerable multimorbidity or polypharmacy are managed in general practice.' This is important and needs emphasis.

Response of the authors: The following has been added under the Conclusions subheading: The implications of this finding are extremely important, as it demonstrates that general practice is bearing most of the burden of increased morbidity and complexity, thereby absorbing excess workload and saving hospital outpatients appointments, emergency department presentations and hospital admissions. Consequently, there are benefits for healthcare economics as well as for the lives of patients who can be managed in the community.

10) Discussion, Strengths and Limitations: the authors state: ' This approach of using multiple sources has been considered advantageous rather than relying on adherence to coding by the GPs'. I suspect the authors don't have the all-important data which would provide a strong justification for their approach. Namely, how much of the data they found were buried away in un-coded data? That would justify manual data searching as in this study, and as opposed to electronic data searching of coded (no freetext) anonymised databases such as CPRD.

Response of the authors: the reviewer is correct on both counts; the proportion of the data found through file-searching of un-coded data would be very helpful and, unfortunately, we did not include the location of recorded information in our protocol.

11) Discussion, Strengths and Limitations: there are many limitations which are not mentioned. A key limitation is how MM was classified (what was and what was not an MM). Why weren't Health Care Assistant consultations included or other healthcare providers within primary care (such as dietitians, midwives, physios etc, if present)? Also, as stated above, what about patients accessing hospital care by bypassing their GP, e.g. emergency admissions - was this data retrospectively captured through inclusion of hospital discharge summaries (this isnt clear in the Methods).

only'

Response of the authors: the authors have added the following sentences to the limitations section to address the points above: 

“…the approach may miss self-presentations to the hospital, especially to emergency departments, and such presentations were not included in this study.”

“Irish general practice software systems are not generally used by other healthcare professionals working in primary care such as physiotherapists, dieticians and healthcare assistants; this study involves GPs and practice nurses only. Finally, this study used a definition of multi-morbidity of two or more chronic conditions, based on a pre-defined list that was provided for the students and set by the clinicians involved in the research team, and conditions not on the list were not included in the dataset.”

Reviewer #2: The study objectively shows what many GPs feel on a daily routine: the workload is increasing, the patient cases are becoming more and more complex. Thus, GPs need new tools to be able to treat patients as efficiently and professionally as possible. In my view, the most important result of their study is the constant rate of referrals to hospital. The protective effect for the clinics should not be underestimated. The results of the study are comprehensible and answer the research question. Essentially, I still miss a more detailed description of the data collected:

Response of the authors: we thank the reviewer for positive feedback and for acknowledging the importance of the results in the context of the role of general practice in healthcare systems. Some of the detail is reported in a previous publication emanating from this dataset (Dinsdale, 2021). We will address the specific feedback below.

What were the characteristics of the participating GP practices? Number of doctors, age, number of patients, etc.

Response of the authors: A new table and the following paragraph have been inserted.

All of the practices were mixed public-private, were computerised and had a practice nurse. Table 1 compares the profile of participating practices to the national profile in 2015 (26). Most of the study practices (93%) had a co-operative system of out-of-hours cover, similar to the national profile (92%). Participating practices had higher percentages involved in postgraduate GP training (43% v 22%) and rural location (37% v 21%). In terms of practice size, 68% had between 500 and 1999 patients; 16% were single-handed practices, 31% had two GPs, 24% had three GPs and 29% had four or more GPs.

Is there information on the most common diseases or medication groups that led to the definition of multimorbidity and polypharmacy?

Only a count of the number of chronic conditions and number of medications was recorded. 

In which specialist departments were the patients referred?

Response of the authors: The specialties most frequently referred to were internal medicine (29.7% of all referrals), emergency department (11.8%), general surgery (10.1%), orthopaedic surgery (8.0%) and medical assessment unit (6.8%). This has been previously reported in Dinsdale E, Hannigan A, O’Connor R, O’Doherty J, Glynn L, Casey M, Hayes P, Kelly D, Cullen W, O’Regan A. Communication between primary and secondary care: deficits and danger. Family practice. 2020 Feb;37(1):63-8.

Are there diseases that led to particularly frequent referrals - where did outpatient care in GP practices reach its medical limits?

Response of the authors: our data cannot tell us which chronic illnesses led to the most referrals. We have reported the frequency of referral to each hospital department previously. 

(Dinsdale E, Hannigan A, O’Connor R, O’Doherty J, Glynn L, Casey M, Hayes P, Kelly D, Cullen W, O’Regan A. Communication between primary and secondary care: deficits and danger. Family practice. 2020 Feb;37(1):63-8.)

Perhaps additional tables could be used to describe the cohort studied even better and thus make it more comparable for other international data!?.

Response of the authors: A new table and the following paragraph have been inserted.

All of the participating practices were mixed public-private, were computerised and had a practice nurse. Table 1 compares the profile of participating practices to the national profile in 2015. Most of the study practices (93%) had a co-operative system of out-of-hours cover, similar to the national profile (92%). Participating practices had higher percentages involved in postgraduate GP training (43% v 22%) and rural location (37% v 21%). In terms of practice size, 75% had between 500 and 1999 patients; 16% were single-handed practices, 31% had two GPs, 24% had three GPs and 29% had four or more GPs.

Note to the reviewers – this study was a retrospective analysis of practice patients and did not distinguish between consultations conducted by various GPs at each practice. We have not, therefore, included gender and age of the GP-tutors as it was not necessarily their own particular consultations (and referrals) that were included.

6. PLOS authors have the option to publish the peer review history of their article (what does this mean?). If published, this will include your full peer review and any attached files.

Do you want your identity to be public for this peer review? For information about this choice, including consent withdrawal, please see our Privacy Policy.

Reviewer #1: Yes: Mark Ashworth

Reviewer #2: Yes: Markus Bleckwenn

---

## [Decision Letter · Decision Letter 1]

28 Dec 2021

PONE-D-21-15158R1How do multi-morbidity and polypharmacy affect general practice attendance and referral rates? A retrospective analysis of consultations.PLOS ONE

Dear Dr. O'Regan,

Thank you for submitting your manuscript to PLOS ONE. After careful consideration, we feel that it has merit but does not fully meet PLOS ONE’s publication criteria as it currently stands. Therefore, we invite you to submit a revised version of the manuscript that addresses the points raised during the review process.

We look forward to receiving your revised manuscript.

Kind regards,

Vijayaprakash Suppiah, PhD

Academic Editor

PLOS ONE

Journal Requirements:

Reviewers' comments:

Reviewer's Responses to Questions

**Comments to the Author**

1. If the authors have adequately addressed your comments raised in a previous round of review and you feel that this manuscript is now acceptable for publication, you may indicate that here to bypass the “Comments to the Author” section, enter your conflict of interest statement in the “Confidential to Editor” section, and submit your "Accept" recommendation.

Reviewer #1: (No Response)

Reviewer #2: All comments have been addressed

2. Is the manuscript technically sound, and do the data support the conclusions?

Reviewer #1: Yes

Reviewer #2: Yes

3. Has the statistical analysis been performed appropriately and rigorously? 

Reviewer #1: Yes

Reviewer #2: Yes

4. Have the authors made all data underlying the findings in their manuscript fully available?

Reviewer #1: No

Reviewer #2: Yes

5. Is the manuscript presented in an intelligible fashion and written in standard English?

Reviewer #1: Yes

Reviewer #2: Yes

6. Review Comments to the Author

Reviewer #1: Thank you for asking me to review the revised version of this paper. The revisions have been substantial and greatly improved the standard of this piece of research. I have minor comments only:

1) Abstract: this paper is in large part about hospital referral rates for patients with multimorbidity. More precision is required in the Results section of the Abstract which simply says: 'Half of patients had been referred to hospital in the previous two years'. The exact % figure should be given to one decimal place.

2) Abstract: a key strength of this work is the use of manual record searching rather than searching of an electronic database. This feature should be added to the Abstract e.g. 'manual record searching'.

3) pg22: minor typo. The sentence says 'aged eligibility' and should read 'age eligibility'. Also, pg 28: 'Analysis of a large clinical practice research database in England reported higher health care utilisation in general practice, higher prescription medications...', should read '...higher number of prescription medications', or similar.

4) A further Limitation to be added is that the authors have not identified which LTCs or medications are associated with increased GP/Practice Nurse/Home Visit attendance rates. So we dont know in clinical terms what the drivers are for increased attendance rates (is it mental health conditions, or diabetes, or hypertension, etc etc, all commonly managed in primary care)?

Reviewer #2: (No Response)

7. PLOS authors have the option to publish the peer review history of their article (what does this mean?). If published, this will include your full peer review and any attached files.

Reviewer #1: No

Reviewer #2: **Yes: **Markus Bleckwenn

---

## [Author Response · Author response to Decision Letter 1]

12 Jan 2022

12/01/2021

Dear editor and reviewers,

The authors are very grateful for the thorough reading and subsequent comments received throughout the review process. The edits from this round of feedback have been helpful, and the paper has been strengthened by them. Please find our point-by-point responses below.

Best wishes,

Dr Andrew O’Regan,

General Practitioner and Senior Lecturer in General Practice

University of Limerick School of Medicine

Review Comments to the Author

Reviewer #1: Thank you for asking me to review the revised version of this paper. The revisions have been substantial and greatly improved the standard of this piece of research. I have minor comments only:

Response of the authors: thank you for this acknowledgement and the comments that follow, all of which are helpful.

1) Abstract: this paper is in large part about hospital referral rates for patients with multimorbidity. More precision is required in the Results section of the Abstract which simply says: 'Half of patients had been referred to hospital in the previous two years'. The exact % figure should be given to one decimal place.

Response of the authors: the results section of the abstract now reads: “50.1% of patients had been referred to hospital in the previous two years.”

2) Abstract: a key strength of this work is the use of manual record searching rather than searching of an electronic database. This feature should be added to the Abstract e.g. 'manual record searching'.

Response of the authors: the following clause has been added to the methods section of the abstract: “Through manual record searching…”

3) pg22: minor typo. The sentence says 'aged eligibility' and should read 'age eligibility'. Also, pg 28: 'Analysis of a large clinical practice research database in England reported higher health care utilisation in general practice, higher prescription medications...', should read '...higher number of prescription medications', or similar.

Response of the authors: the typos on page 22 and again on page 28 have been corrected. Thank you for spotting.

4) A further Limitation to be added is that the authors have not identified which LTCs or medications are associated with increased GP/Practice Nurse/Home Visit attendance rates. So we dont know in clinical terms what the drivers are for increased attendance rates (is it mental health conditions, or diabetes, or hypertension, etc etc, all commonly managed in primary care)?

Response of the authors: We agree that this is an important limitation and have added the following sentence to the limitations section: “Finally, during data collection the number of chronic conditions and medications were recorded for each case but not the name of the chronic condition or medication, and, consequently, the analysis cannot identify which chronic conditions or medications were associated with attendance.”

---

## [Editor Report · Decision Letter 2]

17 Jan 2022

How do multi-morbidity and polypharmacy affect general practice attendance and referral rates? A retrospective analysis of consultations.

PONE-D-21-15158R2

Dear Dr. O'Regan,

We’re pleased to inform you that your manuscript has been judged scientifically suitable for publication and will be formally accepted for publication once it meets all outstanding technical requirements.

Kind regards,

Vijayaprakash Suppiah, PhD

Academic Editor

PLOS ONE

---

## [Editor Report · Acceptance letter]

24 Jan 2022

PONE-D-21-15158R2 

How do multi-morbidity and polypharmacy affect general practice attendance and referral rates? A retrospective analysis of consultations 

Dear Dr. O'Regan:

I'm pleased to inform you that your manuscript has been deemed suitable for publication in PLOS ONE. Congratulations! Your manuscript is now with our production department. 

Kind regards, 

on behalf of

Dr. Vijayaprakash Suppiah 

Academic Editor

PLOS ONE